# Health professions faculty's perceptions of online teaching and learning during the COVID-19 pandemic

**Midion Mapfumo Chidzonga**[1]*, **Clara Haruzivishe**[2], **Vasco Chikwasha**[3], **Judith Rukweza**[4]

1 Department of Oral Health, Faculty of Medicine and Health Sciences, University of Zimbabwe, Harare, Zimbabwe, 2 Department of Primary Health Care, Unit of Nursing Sciences, Faculty of Medicine and Health Sciences, University of Zimbabwe, Harare, Zimbabwe, 3 Department of Primary Health Care, Unit of Global and Public Health, Faculty of Medicine and Health Sciences, University of Zimbabwe, Harare, Zimbabwe, 4 Nursing Sciences, Department Primary Health Care, Faculty of Medicine and Health Sciences, University of Zimbabwe, Harare, Zimbabwe

* mtmchidzonga@yahoo.com

**Data Availability Statement:** All relevant data can be found here: https://data.mendeley.com/datasets/jmgng9dbc3.

## Abstract

The global societal impact of the COVID-19 pandemic is incalculable with profound social suffering, deep economic hardships and enforced closure of schools, businesses, and higher learning institutions through the imposition of lockdown and social distancing in mitigation of the spread of the SARS-Cov-2 infection. Institutions have had to hastily migrate teaching, learning and assessment to online domains, at times with ill-prepared academics, students and institutions and with unwelcome and disorienting consequences. Our study surveyed perspectives of faculty at the University of Zimbabwe Faculty of Medicine and Health Sciences (UZFMHS) towards the hastily adopted online teaching, learning and assessment implemented in response to the mitigation of the COVID-19 pandemic. Twenty nine (29) faculty in all the major disciplines and career hierarchy. There were mixed responses regarding the use of this modality for teaching, learning and assessment: training before online teaching, learning and assessment, advantages and disadvantages, cost effectiveness, effectiveness for teaching, learning and assessment, effect on student feedback, disruptions from internet connectivity issues, interaction with students, suitability for practical training, and barriers to online teaching, learning and assessment. These results would enable the UZFMHS develop institutional and personalised approaches that would enable execution of online teaching, learning and assessment under the current and post COVID-19 pandemic.

## Introduction

The World Health Organisation (WHO) declared COVID-19 a global public health emergency of international concern on 30[th] January 2020 and subsequently declared it a pandemic on 11[th] of March [1, 2]. The societal impact of COVID-19 is incalculable as the pandemic continues to cause profound social suffering and deep economic hardships especially for society's most vulnerable and less fortunate [1–3]. Many aspects of life have been seriously disrupted by

**Funding:** 1.All authors were under the same funding: initials are MMC, COH, JR, VC 2.www.norad.no 3.Funded by the Norwegian Programme for Capacity Development in Higher Education and Research Development (NORHED) The funders had no role in study design, data collection and analysis, decision to publish, or preparation of the manuscript.

**Competing interests:** The authors have declared that no competing interests exist.

the COVID-19 pandemic. The lockdown measures, in a bid to mitigate the spread of the SARS-Cov-2 infection, have led to enforced closure of schools, colleges, and universities, but without cessation of all learning, teaching and assessment although many forms of assessment have had to be suspended [2, 4]. Most institutions of higher learning had to hastily migrate to the online domains with some of them with poor online infrastructures as well as ill-prepared academics and students [5–7]. The new experience has been unwelcome and disorienting to some educators, and students alike. The pandemic has also exposed deficiencies in the use of technology in higher education, which has long been debated, and is now overdue [2]. The University of Zimbabwe Faculty of Medicine and Health Sciences (UZFMHS) now has to embrace this change hastily. This new challenge along with the attendant obstacles has to be faced by many higher education institutions [8]. There is no indication when the pandemic will be contained and as such educational institutions across the globe have now opted to use the available technical resources to create online teaching, learning and assessment material for students of all academic levels [9, 10]. The pandemic has thus created a forced digitalisation change, which was long overdue, and an opportunity to accelerate digital transformation, in health professions education [11, 12]. This is a positive move post COVID-19 pandemic. Pre-clinical health professions education can be taught through webinars. Unfortunately, online teaching, learning, and assessment is most useful for theoretical content but innovative ways have had to be developed to enable clinical training and assessment [13–16]. It is known that formative and high-stakes examinations could be carried out by modifying the current assessment processes such as multiple choice questions, short-answer questions can be done online at the same time viva voce, teleconferencing between candidate and examiners; use of simulation/ objective structured clinical examination; modify patient contact stations to simulated patients; task-trainers and hybrid simulations; patient requiring components; use of personal protective equipment (PPE) at each station; examiner view the examination process remotely. Clinical training depends heavily on hands-on- patient contact to enable clinical skills acquisition, practical work in the laboratories which would require advanced technology which is currently unavailable in most low and middle income countries (LMICs). However, collaboration with high income countries (HIC) may alleviate this situation. Innovative approaches are being put in place to enable continued clinical training: vaccination of all students and faculty; provision of adequate PPE to students, faculty and patients; good hand hygiene; mask wearing; face shields; social distancing; full-body PPE(long-sleeved disposable gown, eye protection, gloves) for aerosol producing procedures; and small numbers of students at a time. For non-aerosol producing procedures, minimum PPE is provided (disposable apron, facemask, gloves and eye protection is provided. This ensures that the safety of patients, students and staff is not compromised.

Health professions students need to be exposed to patients in an authentic environment, an apprenticeship type of learning. This is usually in hospital wards, outpatient clinics and simulated teaching laboratories. Under the current COVID-19 pandemic live broadcasts, films or virtual case presentations, and surgical procedures might be used to complement the limited access to patients.

Faculty are expected to increase their digital competence so as to come up with innovative virtual teaching, learning, and assessment tools.

The COVID-19 pandemic has curtailed clinical placements as well as elective services offered in clinical services. However, there is need to continue to adequately train health professionals to run the healthcare system. Student clinical placement has to continue while observing safe supervision of students, adequate provision of PPE, social distancing, washing and sanitising hands. Various learning management systems are available: Moodle, Massive Open Online Courses. Several tools are also available that allow for video communications,

video and audio conferencing, chats, and webinars: Skype, Zoom, and Google Hangouts Meet, Google school [13]. It is possible to set up real-time classroom experience that simulates patient encounters using live HD-quality mobile or fixed-cameras that ensures that all distance learners can collaborate at an equivalent baseline [14]. Current virtual learning management systems offer access to educational content from anywhere, synchronously or asynchronously with interactive simulation learning [13]. Unfortunately the theoretical knowledge has the challenge of not being able to be transferred to patient management. The way forward would be to introduce interactive simulation learning (computerised simulation education) which unfortunately need massive infrastructure ranging from clinical simulation management software and hardware, design and planning tools, file backup, cloud-based eLearning, and expert teams to provide support to counsellor education, case developers, and virtual patient training [13].

This study at the UZFMHS surveyed the faculty's perspectives towards online teaching, learning and assessment. The information gathered would enable development of online approaches tailor made based the perceptions of faculty.

## Methodology

### Setting

The study was conducted at the UZFMHS which provides health professions education and training to various healthcare cadres.

### Study design

Mixed method design.

### Ethical considerations

Permission to carry out the study was obtained from the Dean of the UZFMHS. Ethical clearance was obtained from the local ethical review board, Joint Research Ethics Committee of the UZFMHS and Parirenyatwa Group of Hospital (JREC) and the Medical Research Council of Zimbabwe (MRCZ). Informed consent was obtained from the participants. There were no risks to the participants. Participation was voluntary. A written consent form was sent to the participants online. Participants were requested to read the consent form and only proceed with the questionnaire if they were agreeable with the information given on the consent form. The study did not include minors. Benefits to the participants may accrue should their perspectives be used to enrich the development of online teaching, learning and assessment.

### Study participants

The participants were faculty of medicine, dentistry, nursing, physiotherapy, occupational therapy, pharmacy, health promotion and education, radiographers and medical laboratory scientists.

### Sampling

We invited faculty in the above disciplines to participate.

### Study instruments

An online questionnaire was created in Google Forms and emailed to faculty. The questionnaire captured demographics, academic grade, number of years in teaching, computer literacy,

**Table 1. Shows answers to questions regarding online teaching and learning.**

| QUESTIONS | YES | NO |
|---|---|---|
| | %/N | %/N |
| Have you received training on online teaching and learning? | 65.5%/19 | 34.5%/10 |
| Do you enjoy online teaching and learning? | 48.3%/14 | 51.7%/15 |
| Do you believe in benefits of online learning to students? | 93%/27 | 7%/2 |
| Do you believe that online learning enhances your teaching capacity | 48%/14 | 52%/15 |

device used when delivering teaching type of internet connections used, whether one received training on online teaching and learning, benefits of online learning to students, role of online learning in enhancing teaching, belief in online teaching, cost effectiveness, convenience for teaching, disadvantages and advantages of online education, barriers to online teaching, and online teaching practice, and platforms used in teaching.

## Data analysis

Data collected through the online survey was exported to STATA 15.1 statistical software for cleaning, analysis, calculation frequencies and percentages for categorical data. Qualitative data is presented as themes and subthemes. Frequency responses to "YES" and "NO" items are presented in the tables.

## Results

Twenty nine (29) faculty responded, 62.1% (n = 18) males and 37.9% (n = 11) females. The teaching experience of faculty ranged from 2 years to 44 years. Sixty five point five per cent (65.5%, n = 19) had received training on online teaching and 34.5% had not received any training.

On belief in online teaching, Table 1, shows responses to "Yes and "No" questions regarding online teaching, learning and assessment with 65.5% of the respondents indicating that they had received training on online teaching, learning and assessment indicative of a possible positive participation in online teaching. However, 51.7% did not enjoy teaching online with 52.0% believing that online teaching does not enhance their teaching capacity.

Table 2. Shows the academic grades of faculty. Teaching experience of faculty ranged from 2 to 44 years. All the faculty members were comfortable using a computer for teaching and learning. The majority of the faculty respondents are in the lecturer and senior lecturer grade. This is a positive situation as this group is likely to be in the age group that is technology savvy and should be able to navigate the education technology with ease.

Table 3. Shows the devices used in delivery of teaching. The majority of faculty, 82.7%, deliver their lectures using laptops which means they could deliver their teaching at any location where there is internet connectivity.

**Table 2. Shows academic grade of faculty.**

| Grade | N (%) |
|---|---|
| Professor | 8 (27.6%) |
| Associate Professor | 0 (0.0%) |
| Senior Lecturer | 10 (34.5%) |
| Lecturer | 11 (37.9%) |
| TOTAL | 29 (100%) |

**Table 3. Shows devices for delivering teaching.**

| Device | N (%) |
|---|---|
| Smartphone | 0 (0%) |
| Tablet | 1 (3.5%) |
| Laptop | 24 (82.7%) |
| Personal Computer | 3 (10.3%) |
| Other (device not indicated) | 1 (3.5%) |
| Total | 29 (100.0%) |

**Table 4. Shows type of internet connection used.**

| Internet connection used | N (%) |
|---|---|
| Mobile network | I (3.5%) |
| W-LAN (wireless connection) | 15 (51.7%) |
| LAN (cable connection) | 10 (34.5%) |
| Both WLAN and LAN | 3 (10.3%) |
| Total | 29 (100%) |

**Table 5. Shows themes and subthemes for those who enjoyed online teaching in response to the question whether one enjoyed online teaching.**

| Number | Theme | Subtheme |
|---|---|---|
| 1 | Convenient for teaching | It is a convenient method |
| | | It is flexible |
| | | Effective with big numbers |
| | | More practical because one doesn't have to travel to teach |
| | | I can teach from my office without having to go to different lecture rooms for different year groups |
| | | It is convenient as I can continue with my teaching even when I am not physically in the country |
| | | Teaching can be done at a convenient time |
| | | It is suited for a busy clinician |
| 2 | Cost effective | It is cheaper |
| | | It cuts on transport cost |
| | | A lecture can be recorded for future us |
| | | Challenges are welcome |
| 3 | More effective teaching methods explored and applied | Gives room to explore other ways of teaching like simulation and videos |
| | | Helps facilitator simplify teaching and produce quality lecture presentations. |
| | | It is easier to attach videos and pictures to illustrate a point. |
| 4 | . Safer in the prevailing COVID-19 pandemic | It is safer especially in this prevailing COVID-19 pandemic. |
| | | A novel method. |

Table 4. Shows that most (51.7%) faculty received their internet connection through W-LAN (wireless connection, local area network), followed (34.5%) by the LAN (connection). This enables them to conduct their teaching wherever there is wireless internet connectivity.

Table 5. Shows themes and subthemes for those who enjoyed online teaching (48.3%, n = 14) in response to the question whether one enjoyed online teaching along with

**Table 6. Shows themes and subthemes for those who indicated that they did not enjoy online teaching in response to the question whether one enjoyed teaching online.**

| Number | Theme | Sub-theme |
|---|---|---|
| 1 | Lack of physical interaction with students | I am comfortable with my students |
| | | It lacks interaction |
| | | You cannot see students as you teach so you don't know if what you deliver is effective and understood |
| | | I prefer physical interaction with students |
| | | Less interaction with students |
| | | I teach better with face–to- face teaching |
| | | I do not link with 60% of my students as they depend on other students as they re-route information via other platforms like WhatsApp |
| | | I cannot see students as they switch off their videos |
| 2 | Frequent disruption of lecture due to network challenges | Internet connectivity not good |
| | | This is a whole new experience which is marred by lack of suitable internet connectivity, lack of data and resources. |
| | | Network challenges can be a problem |
| | | I often have internet problems leading to disruption in teaching |
| 3 | Lack of immediate feedback | Student facial response and engagement allows me to continuously modify my teaching |
| | | It is not possible to assess facial expressions to see if students understood. |
| | | There is lack of immediate feedback. |
| 4 | Unsuitable for practical based learning | Online learning is unsuited for practical based learning |

disadvantages and advantages of online teaching, convenient for teaching, cost effective, and safe teaching modality under the COVID-19 pandemic.

Table 6. Shows the themes and subthemes for those who indicated that they did not enjoy online teaching (51.7%, n = 15) in response to the question whether one enjoyed teaching online. They enumerated some of the disadvantages of online teaching such as clinical teaching and difficulty to give timeous feedback to students.

Eighty three per cent (83%, n = 24) of faculty had commenced online teaching and learning with 17% (n = 5) yet to commence teaching online.

Table 7, shows the platforms in use for online teaching. The most commonly used platform is ZOOM (62.1%, n = 18).

Table 8. Shows barriers to online teaching, learning and assessment and possible solutions. Internet connectivity, high data costs, and inability to pass on technical skills were some of the

**Table 7. Platforms used for online teaching.**

| Platform | N/% |
|---|---|
| Zoom | 18/62.1% |
| Moodle | 5/17.2% |
| E-Mhare | 2/6.9% |
| Google meet | 1/3.4% |
| Google classes | 1/3.4% |
| Zoom and Google | 1/3.4% |
| WhatsApp and emails | 1/3.4% |
| Totals | 29/100% |

**Table 8. Shows the barriers and possible solutions to online teaching and learning.**

| Barrier | Possible Solutions |
| --- | --- |
| 1. University internet connectivity may be unavailable, disrupted or broken down | Lecturers and students should be supported financially so that they secure data for efficient uninterrupted flow of lectures. |
| 2. Non-availability of allowances for internet for lecturer | Provide data or allowance for internet to lecturers to enable smooth lecture planning and teaching off campus. |
| 3. Unable to pass on technical skills as bedside teaching is impossible | To accommodate physical lectures in wards in turn with online teaching. |
| 4. Data costs too high if venue of teaching is not the workplace. | To provide lecturers with internet allowance. |
| 5. Lectures disrupt by use of the limited 40 minute zoom session and time wasted when re-logging in. | The institution to subscribe to the unlimited facility. |
| 6. Unavailability of internet equipment in the units | The purchase of internet equipment by the institution |
| Inattention by students | More training is required about the monitoring of student attendance and participation. |

barriers to online teaching. Solutions offered included a subsidy to faculty to access data and incorporate facilities that would beam online activities in the wards.

## Discussions

The lockdowns and subsequent closure of higher education institutions are mitigation measures to the spread of the SARS-CoV-2 virus and protect the students, faculty and the community but at the same time disrupted the training of future healthcare professionals due to loss of time of learning/training and inability to continue adequate clinical training [15]. The COVID-19 pandemic has caused the closure of university campuses around the world and migration of all learning, teaching, and assessment to online domains [2]. This has caused profound changes in health professions education with regards to clinical placements along with the social suffering and extensive economic hardships and strain on healthcare systems. High income countries (HIC) managed to swiftly migrate to e-learning with ease as they already had invested in digitalisation unlike the low and middle income countries (LMIC) that had minimal capacity and require extensive adjustments to achieve the abrupt switch to e-learning [1, 17–22]. The e-learning technology has not been evenly dispersed throughout all nations and cultures [22]. Limited institutional capacity and knowledge on online teaching inevitably made this transition more challenging for LMIC. The HIC successfully swiftly implemented a vast array of technology-enhanced learning solutions within the pre-existing structures: webinars conducted via zoom, Skype, Google Hangouts, WebX; online learning platforms; mobile applications; 3-dimensional anatomy models; and online question banks [19, 23, 24]. In some LMIC distance e-learning had not been adopted as a modality of teaching within health professions education [19]. The availability of essential infrastructures, well-trained educators, advanced technologies, efficient institutional strategies and student related barriers represent major challenges in integrating distance e-learning in health professions education in LMIC [23–26]. With the uncertainty of returning to normal life after the containment of the pandemic e-learning will remain the modality of teaching, learning and assessment in higher education.

Digital learning is a process of integrating technology-mediated synchronous and asynchronous approaches including assessments, assignments, and tutoring enabling learning without any time and location restrictions [15, 27, 28]. Digital learning has a few components including digital teaching materials, digital tools, digital delivery, and autonomous learning.

Health professions education has adopted digital learning with through virtual course, simulation software and teleconferencing and are struggling to come up with suitable virtual teaching and learning for the clinical component of their training that requires the traditional patient contact in the hospital/clinic set up.

This is in general agreement that online learning is a good option for acquiring theoretical knowledge since it is not possible to hold face-to-face learning under the COVID-19 pandemic. Efficacy of online learning has long been acknowledged [29–34]. Hugenholtzel found that e-learning is just as effective in enhancing knowledge as lecture-based learning [31]. A study indicated that online learning was less effective in building skills and knowledge mainly due to the difficulty of interacting with peers and faculty along with the acquisition of skills which are mostly better acquired through in-person interaction [7]. However, literature indicates that high level of satisfaction are related to well -structured and organised e-courses which also have a greater impact on knowledge accumulation and student performance compared with traditional learning [7]. Systematic reviews have reported that blended learning has better effects on knowledge outcomes compared to traditional learning [33, 35, 36].

Blended learning has been adopted by our institution with small groups of students attending clinical settings and in-person lectures intermixed with online learning.

In the current study the majority of faculty (65.5%) had received training in online teaching, learning and assessment. In line with other studies in the literature the technological skills to provide online courses increases the educational value of faculty [37]. There is need to continue to support faculty as they implement e-learning.

Twenty nine (29) lecturers participated in the study: 62.1% (n = 18) were males and 37.9% were females. The majority of the respondents were in the lecturer (37.9%) and senior lecturer grade (34.5%). This could reflect a relatively low experience in teaching and possibly age hence their perception of e-learning is likely to be positive based on their high technology capacity. This could also explain their enjoyment of online teaching, learning and assessment. Fischer et al. stated that older staff with long traditional teaching experience usually have limited interaction with technology and lacking the development of their necessary skills [38, 39].

The most commonly used device for the delivery of teaching was the laptop (82.7%) followed by the personal computer (10.3%). The institution would then be expected to assist faculty with replacement of laptops as support for the continued online teaching, learning and assessment.

Faculty most commonly used W-LAN (51.7%) followed by the LAN (34.5%). The institution would be expected to improve the internet connectivity at the institution and also assist faculty with data when they are away from the office so that they could continue with online teaching wherever they would be especially with the lockdown measures.

In agreement with other studies in the literature the majority of respondents (93.1%) believed that there were benefits in online teaching [37, 38]. Literature indicates that some of the factors among faculty that influence the level of satisfaction of online teaching are self-gratification, intellectual challenge, interest in using technology, and associated professional development opportunities [5, 40]. This study found that the benefits of e-learning were: convenience for teaching, cost effectiveness, effective teaching approach and a safer teaching approach under the COVID-19 pandemic. Most institutions were using blended learning before the advent of the COVID-19 pandemic and did find the switch to completely online overwhelming.

Six point nine percent (6.9%) did not agree that there were any additional benefits compared with face-to-face teaching and that immediate feedback is curtailed, lack of physical interaction with students, network challenges affecting lecture delivery and suitability for practical-based learning, and negative impact on hands on training or skills transfer. They also implored for the need to hold regular training courses on online teaching and learning. This is

in agreement with other studies in the literature that showed that some faculty were not fully comfortable with e-learning as a teaching tool and attributed this to factors such as technological challenges, difficulty in interactions and discussions with students, lack of adequate internet connectivity, students access to internet was difficult and expensive, and personal learning preference [37, 38]. There is need to cultivate a culture of change as institutions transition to online teaching, learning and assessment as this will reduce resistance. Advanced planning and infrastructure are important for successful implementation of e-learning [41]. There is also need to modulate the traditional methods of teaching, learning and assessment for e-learning.

Ninety three per cent (93.0%; n = 27) reported that they believed in the benefits of online teaching of students with 7% (n = 2) indicating that there were no benefits to students accruing from online teaching. The majority of faculty agreed that online teaching enhanced their teaching capacity while those who did not believe in the enhancement from online teaching felt that it added no value to the learning of the students. They indicated the need for further training and adequate appropriate tools and reliable internet connectivity.

The majority (83%; n = 24) are already participating in online teaching with zoom being the most common platform used (62.1%) with social media and email being used to communicate with students. Other platforms were also used: Google meet, Google lessons, e-Mhare. In our faculty both synchronous (live or in real-time) and asynchronous (recorded or self-paced) are used through the University's learning management system (along with Zoom or Microsoft Teams). Synchronous e-learning is offered in the form of interactive teaching and clinical case discussions in small and large group formats [37] and asynchronous e-learning includes preparation of course materials for students in advances to be accessed later (recorded lectures, supportive videos, external links for recommended websites, and electronic books [37, 41].

Literature indicates that there are many significant barriers to the adoption and implementation of e-learning by medical schools [5, 10]. These barriers include technology/infrastructure barriers, institutional/educators' and student barriers [19]. In agreement with other studies from LMIC lack of infrastructure, advanced technology, internet access, and poor quality of internet services (internet band width and connectivity limitations), limited technical support, unfamiliarity with e-learning systems were mentioned as barriers to the transition to virtual teaching/training and learning [15, 19, 20]: unavailability of university internet connectivity (unavailable, disrupted, broken down, interrupted power supply), absence of financial allowances to pay for internet service data (high cost), inattention by students, inability to perform skills transfer, disturbances from the home environment (working at home due to lockdown measures). The disturbances at home are similar to those in other studies which showed that the learning environment at home posed the greatest challenge while technological literacy and competency was the least of their challenges [34]. Low household income in LIMC makes it difficult for one to access the necessary equipment for online learning, along with poor power supply, poor internet connectivity, sociocultural dynamics as gender roles, educators' and learners' competence related to online teaching.

Blended teaching and learning, combining online and face-to-face teaching, balances benefits and drawbacks of online and face-to-face teaching [30]. Distance online course can prepare students as well as face-to-face. However, performance in online learning is reduced due to the COVID-19 driven social distancing and physical isolation as activities such as workshops, laboratories, and clinical rotations have had to be suspended [30].

The teaching environment poses challenges met by faculty inclusive of issues such as distractions at home (e.g., visitors, noise, and home chores) and limited space and facilities [34]. Poor internet connectivity, poor picture quality of videos and poor performing platforms were noted as possible barriers [25]. Students and faculty need continuing training on online

teaching and learning. A needs assessment should be performed among the students, health professions educators and the institution so that the identified needs and quality control be implemented at commencement of online teaching [35, 36, 42]. Poor internet connectivity, poor picture quality of videos and poor performing platforms were noted as barriers in this study as was noted in other studies [25]. The needs assessment would help in the development of appropriate policies and a supportive culture.

## Conclusions

The impact of the emergency embarking on online learning is not yet known. Early stages yet to assess the role of the academics on the future of e-learning in global health professions education. These findings can be used to improve health professions faculty approach to online teaching and learning as the global COVID-19 pandemic appears to be with us for a long time to come.

Institutions should improve the online teaching and learning infrastructures: learning management system (webinar facilities (zoom etc.) online learning platforms, mobile applications, Skype, Google Hangouts, training of both students and faculty on online teaching and learning, simulation/skills, laboratories, develop online curriculum, develop online examinations, telehealth, online video libraries. The developments should be embarked upon in line with their unique challenges of their faculty and the institutional existing online infrastructures. Continuing training and support on e-learning for faculty is important for successful acceptance and implementation of e-learning.

Methods of introducing and supporting health professions faculty transitioning to new teaching and learning circumstances and environments should provide contextualised both online and in-person strategies. Consideration of the negative and positive aspects of e-learning should allow for the development of a blended learning approach, flipped class, team-based learning approach that integrates aspects of face-to-face learning and e-learning to achieve the intended learning outcomes especially in practical and clinical training.

## Author Contributions

**Conceptualization:** Midion Mapfumo Chidzonga, Clara Haruzivishe.

**Data curation:** Midion Mapfumo Chidzonga, Clara Haruzivishe, Vasco Chikwasha, Judith Rukweza.

**Formal analysis:** Midion Mapfumo Chidzonga, Clara Haruzivishe, Vasco Chikwasha, Judith Rukweza.

**Funding acquisition:** Midion Mapfumo Chidzonga.

**Investigation:** Midion Mapfumo Chidzonga, Clara Haruzivishe, Judith Rukweza.

**Methodology:** Midion Mapfumo Chidzonga, Clara Haruzivishe.

**Project administration:** Midion Mapfumo Chidzonga, Clara Haruzivishe, Judith Rukweza.

**Resources:** Midion Mapfumo Chidzonga.

**Software:** Midion Mapfumo Chidzonga, Vasco Chikwasha.

**Supervision:** Midion Mapfumo Chidzonga, Judith Rukweza.

**Validation:** Midion Mapfumo Chidzonga, Vasco Chikwasha.

**Visualization:** Midion Mapfumo Chidzonga.

**Writing – original draft:** Midion Mapfumo Chidzonga, Clara Haruzivishe, Vasco Chikwasha, Judith Rukweza.

**Writing – review & editing:** Midion Mapfumo Chidzonga, Clara Haruzivishe, Judith Rukweza.

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
