## [Decision Letter · Decision Letter 0]

11 Apr 2022

PONE-D-21-33695Health professions faculty’s perceptions of online teaching and learning during the COVID-19 pandemicPLOS ONE

Dear Dr. Chidzonga,

Thank you for submitting your manuscript to PLOS ONE. After careful consideration, we feel that it has merit but does not fully meet PLOS ONE’s publication criteria as it currently stands. Therefore, we invite you to submit a revised version of the manuscript that addresses the points raised during the review process. Also Kindly mention how the questionnaire was developed. Was it validated? Was a pilot testing done? How exactly was the qualitative data analysed?  Please submit your revised manuscript by May 26 2022 11:59PM. If you will need more time than this to complete your revisions, please reply to this message or contact the journal office at plosone@plos.org. Please include the following items when submitting your revised manuscript:A rebuttal letter that responds to each point raised by the academic editor and reviewer(s). You should upload this letter as a separate file labeled 'Response to Reviewers'.A marked-up copy of your manuscript that highlights changes made to the original version. You should upload this as a separate file labeled 'Revised Manuscript with Track Changes'.An unmarked version of your revised paper without tracked changes. You should upload this as a separate file labeled 'Manuscript'.

We look forward to receiving your revised manuscript.

Kind regards,

Pathiyil Ravi Shankar

Academic Editor

PLOS ONE

“Many thanks to the NORHED for funding this project.”

“1.All authors were under the same funding: initials are MMC, COH, JR, VC

2.www.norad.no

3.Funded by the Norwegian Programme for Capacity Development in Higher Education and Research Development

(NORHED)

4.NO.The funders had no role in study design, data collection and analysis, decision to publish, or preparation of the manuscript”

“NO. The authors have declared that no competing interests exist.”

5. PLOS requires an ORCID iD for the corresponding author in Editorial Manager on papers submitted after December 6th, 2016. Please ensure that you have an ORCID iD and that it is validated in Editorial Manager. To do this, go to ‘Update my Information’ (in the upper left-hand corner of the main menu), and click on the Fetch/Validate link next to the ORCID field. This will take you to the ORCID site and allow you to create a new iD or authenticate a pre-existing iD in Editorial Manager. Please see the following video for instructions on linking an ORCID iD to your Editorial Manager account: https://www.youtube.com/watch?v=_xcclfuvtxQ.

Reviewers' comments:

Reviewer's Responses to Questions

**Comments to the Author**

1. Is the manuscript technically sound, and do the data support the conclusions?

Reviewer #1: Partly

Reviewer #2: Yes

Reviewer #3: Partly

Reviewer #4: No

2. Has the statistical analysis been performed appropriately and rigorously? 

Reviewer #1: No

Reviewer #2: Yes

Reviewer #3: Yes

Reviewer #4: No

3. Have the authors made all data underlying the findings in their manuscript fully available?

Reviewer #1: No

Reviewer #2: Yes

Reviewer #3: Yes

Reviewer #4: Yes

4. Is the manuscript presented in an intelligible fashion and written in standard English?

Reviewer #1: No

Reviewer #2: Yes

Reviewer #3: Yes

Reviewer #4: No

5. Review Comments to the Author

Reviewer #1: The Manuscript titled "Health professions faculty’s perceptions of online teaching and learning during the COVID-19 pandemic' gives a different perspective of challenges faced during the pandemic in the context of a low Income Country like Zimbabwe,Africa. The methodology is sound and robust.However,there are a number of things which can be improved in the manuscript.These are as follows:

1.Incomplete sentence in the abstract.

2. Naming of the tools used for teaching and learning correctly in the introduction section.For eg.Google Meet which was earlier called as Hangouts. Need clarification on what is Google School.

3.No mention of whether the questionnaire used in the study was validated prior to it being administered to the faculty.Also suggest to attach the Questionnaire as supplementary material.

4.The results were presented in tables with a legend included.However,the tables could be presented neatly after some more formatting.Table 7 was repeated twice in the manuscript.The written text had a repetition of the table legends and this can be improved.Data presented did not include parameters like Mean,Median and the statistical significance.This can be improved by doing a more rigorous statistical analysis.

5.The discussion section has been written well though occasionally there are repetition of statements.

6.The conclusion section had some incomplete sentences and were grammatically incorrect.This needs to be checked.

7,Many of the references need formatting according to the Journal requirements.Some references had the names of the Journal missing and some did not have the Year,Volume,Issue and Page numbers.This needs to be checked using a reference manager like EndNote or Mendeley.

8.Overall,there were numerous grammatical errors and missing punctuation markers.I would recommend the manuscript be proofread by a native English speaker or the manuscript be sent to a Professional Proofreading Service to amend the language used in the manuscript.This would ensure that the manuscript would be palatable to readers all over the world.

The manuscript can be considered for publication if all these issues are addressed.

Reviewer #2: Authors have put their good efforts in doing this study and writing manuscript. Authors have to clarify the queries and update the manuscript.

Abstract

it is unstructured. Authors have just mentioned the mixed response of participants on all variables; it will be better if they can elaborate important findings with data.

Methodology

Authors have to mention the study period and about piloting of instrument.

Results

Authors have to mention response rate of faculty. Even though the authors have given sequence to Tables but after table 1, table 7 appears; this could be due to mistake and needs to be corrected.

Discussion

First four paragraphs seems to be review of literature; this can be added in introduction. Discussion must focus on comparing and contrasting the findings with other study and inference of the findings.

Authors have to mention the limitation of study.

Conclusion

Conclusion should be drawn from findings of study. Authors have to look into and rewrite the conclusion.

Reviewer #3: 1.The topic for study is relevant to the current situation faced by faculty of higher education.

2. Please describe the sample selection process.

3. There are many repetitions in the introduction and discussion sections as well as in ethical clearnace part, please review and revise it.

4. Methodology section needs to be revised.

Reviewer #4: The topic is interesting.

The introduction would benefit from synthesis/integration of prior research into Health professions faculty’s perceptions of online teaching and learning during the COVID-19 pandemic globally in general and Africa in particular. Identify the research gap followed by the need of the study.

Methodology:

The study design is a mixed method design. However, the methods in the main body lack detail and do not reflect an understanding of qualitative research. The plan for analysis should contain more detail. Which method of thematic analysis was followed? Who coded the data? Was it an iterative process between authors? How were themes determined?

Quantitative data: Sample Size is 29. The data was collected by online Google form which was emailed to the faculty. Taking into account that the participants were faculty of medicine, dentistry, nursing, physiotherapy, occupational therapy, pharmacy, health promotion and education, radiographers and medical laboratory scientists, the sample size is not adequate.

It is not mentioned in the paper anywhere when the study was conducted. (Year/ month of study).

Sampling: Which technique was used?

Results:

Table 7. Platforms used for online teaching have been repeated twice.

Table 8: About the possible solutions to the barriers to online teaching, whether the data was collected from the participants? as it seems to be generated by the authors.

Discussion: It has to be revised. Major part of the discussion is just the repetition of results.

Few of the sentences are not clear as authors need to specify which study they are referring to.

Eg. In agreement with other studies in the literature…….be more specific

However, literature indicates…….?

Literature indicates that some of the factors…..?

In line with other studies in the literature ……..?

Literature indicates……..authors need to be more specific about which study they are referring to and was the finding of the study.

Conclusions

It is not drawn from the findings of the study.

6. PLOS authors have the option to publish the peer review history of their article (what does this mean?). If published, this will include your full peer review and any attached files.

Reviewer #1: No

Reviewer #2: **Yes: **Rano Mal Piryani

Reviewer #3: **Yes: **Dr. Medha Anant Joshi

Reviewer #4: **Yes: **Indrajit Banerjee

---

## [Author Response · Author response to Decision Letter 0]

23 May 2022

Faculty research Response to First reviewer comments

1. How many were invited? Give the number please.

We used a convenient sample of 100 faculty from the above disciplines. The faculty approached where those who were accessible on campus during the time of data collection. Only 29 faculty responded.

2. was the questionnaire validated? Totally how many questions were there related to the demographic information and how many related to online teaching and learning?

The tool used to collect data was piloted on 5 faculty who were excluded from the main study. Following this pilot some items were removed and only relevant items were left: three 3 questions relating to demographic data and 13 questions relating to online teaching and learning were finally used for the study.

3. This has already been described in the introduction.

This has already been described in the introduction.

4. what is the relevance of this to the study carried out?

This will be removed. 

5. What percentage of the total faculty members participated?

we got responses from 29 out of 100 members approached.

6. It is saying the same thing as the previous paragraph

This will be removed

7. As per the table 1, 48% agreed while 52% disagreed, that makes this presumption of "majority" of the faculty, invalid

This has been rephrased accordingly: The majority of faculty did not agreed that online teaching enhanced their teaching capacity while those who believed in the enhancement from online teaching felt that it added value to the learning of the students

 

Reviewers comments from second reviewer

1. How many were invited? Give the number please.

a. 100 faculty were invited 

2. was the questionnaire validated? Totally how many questions were there related to the demographic information and how many related to online teaching and learning?

a. Yes. The questionnaires were developed using relevant items that were used in literature. The tool was then piloted on 5 faculty who were excluded from the main study. Following this pilot some items were removed and only relevant items were left: three 3 questions relating to demographic data and 13 questions relating to online teaching and learning were finally used for the study.

3. This has already been described in the introduction.

a. Agreed. this has been removed.

4. what is the relevance of this to the study carried out?

a. Nor relevant. This has been removed.

5. What percentage of the total faculty members participated?

a. 29 out of 100

6. It is saying the same thing as the previous paragraph

a. This will be removed 

7. As per the table 1, 48% agreed while 52% disagreed, that makes this presumption of "majority" of the faculty, invalid

a. In agreement. This has been rephrased as follows: "The majority of faculty did not agree that on-line teaching enhanced their teaching capacity while those who believed in the enhancement from on-line teaching felt that it added value to the learning of the students. "

---

## [Decision Letter · Decision Letter 1]

14 Jun 2022

PONE-D-21-33695R1Health professions faculty’s perceptions of online teaching and learning during the COVID-19 pandemicPLOS ONE

Dear Dr. Chidzonga,

Thank you for submitting your manuscript to PLOS ONE. After careful consideration, we feel that it has merit but does not fully meet PLOS ONE’s publication criteria as it currently stands. Therefore, we invite you to submit a revised version of the manuscript that addresses the points raised during the review process.

Kindly revise as per the reviewers comments.

We look forward to receiving your revised manuscript.

Kind regards,

Pathiyil Ravi Shankar

Academic Editor

PLOS ONE

Journal Requirements:

Reviewers' comments:

Reviewer's Responses to Questions

**Comments to the Author**

1. If the authors have adequately addressed your comments raised in a previous round of review and you feel that this manuscript is now acceptable for publication, you may indicate that here to bypass the “Comments to the Author” section, enter your conflict of interest statement in the “Confidential to Editor” section, and submit your "Accept" recommendation.

Reviewer #1: All comments have been addressed

Reviewer #4: (No Response)

2. Is the manuscript technically sound, and do the data support the conclusions?

Reviewer #1: Yes

Reviewer #4: Partly

3. Has the statistical analysis been performed appropriately and rigorously? 

Reviewer #1: Yes

Reviewer #4: Yes

4. Have the authors made all data underlying the findings in their manuscript fully available?

Reviewer #1: Yes

Reviewer #4: Yes

5. Is the manuscript presented in an intelligible fashion and written in standard English?

Reviewer #1: Yes

Reviewer #4: Yes

6. Review Comments to the Author

Reviewer #1: I believe that most of my comments have been addressed.There are some minor errors in the results section which need to be corrected in order for the article to be accepted for publication.

Reviewer #4: There are 4 peer reviewers for this paper. I am not able to find response from the authors to the 4 reviewers. I can only find response to reviewer 1 and the same text has been repeated for the response to the reviewer 2.

7. PLOS authors have the option to publish the peer review history of their article (what does this mean?). If published, this will include your full peer review and any attached files.

Reviewer #1: **Yes: **Dr Sunil Pazhayanur Venkateswaran

Reviewer #4: **Yes: **Indrajit Banerjee

---

## [Author Response · Author response to Decision Letter 1]

24 Sep 2022

We have now submitted the original document with track changes and a clean Manuscript after accepting track changes

---

## [Decision Letter · Decision Letter 2]

2 Oct 2022

Health professions faculty’s perceptions of online teaching and learning during the COVID-19 pandemic

PONE-D-21-33695R2

Dear Dr. Chidzonga,

We’re pleased to inform you that your manuscript has been judged scientifically suitable for publication and will be formally accepted for publication once it meets all outstanding technical requirements.

Kind regards,

Pathiyil Ravi Shankar

Academic Editor

PLOS ONE

Additional Editor Comments (optional):

Reviewers' comments:

Reviewer's Responses to Questions

**Comments to the Author**

1. If the authors have adequately addressed your comments raised in a previous round of review and you feel that this manuscript is now acceptable for publication, you may indicate that here to bypass the “Comments to the Author” section, enter your conflict of interest statement in the “Confidential to Editor” section, and submit your "Accept" recommendation.

Reviewer #1: All comments have been addressed

Reviewer #2: All comments have been addressed

2. Is the manuscript technically sound, and do the data support the conclusions?

Reviewer #1: Yes

Reviewer #2: Yes

3. Has the statistical analysis been performed appropriately and rigorously? 

Reviewer #1: Yes

Reviewer #2: Yes

4. Have the authors made all data underlying the findings in their manuscript fully available?

Reviewer #1: Yes

Reviewer #2: Yes

5. Is the manuscript presented in an intelligible fashion and written in standard English?

Reviewer #1: Yes

Reviewer #2: Yes

6. Review Comments to the Author

Reviewer #1: Perhaps to include a statement on the limitations of the study keeping in mind that only small numbers(n=29) responded to the online survey.Other points to consider is that students and faculty are now able to return to their institutions as more face to face sessions have returned and there is a hybrid arrangement for teaching and learning in most institutions.

Reviewer #2: (No Response)

7. PLOS authors have the option to publish the peer review history of their article (what does this mean?). If published, this will include your full peer review and any attached files.

Reviewer #1: **Yes: **Sunil Pazhayanur Venkateswaran

Reviewer #2: **Yes: **RANO MAL Piryani
